# Food Bioactive Compounds and Their Interference in Drug Pharmacokinetic/Pharmacodynamic Profiles

**DOI:** 10.3390/pharmaceutics10040277

**Published:** 2018-12-14

**Authors:** Matteo Briguglio, Silvana Hrelia, Marco Malaguti, Loredana Serpe, Roberto Canaparo, Bernardo Dell’Osso, Roberta Galentino, Sara De Michele, Carlotta Zanaboni Dina, Mauro Porta, Giuseppe Banfi

**Affiliations:** 1IRCCS Orthopedic Institute Galeazzi, Scientific Direction, 20161 Milan, Italy; banfi.giuseppe@fondazionesanraffaele.it; 2Department of Life Quality Studies, University of Bologna, 47921 Rimini, Italy; silvana.hrelia@unibo.it (S.H.); marco.malaguti@unibo.it (M.M.); 3Department of Drug Science and Technology, University of Torino, 10125 Torino, Italy; loredana.serpe@unito.it (L.S.); roberto.canaparo@unito.it (R.C.); 4Department of Biomedical and Clinical Sciences Luigi Sacco, Department of Psychiatry, ASST Fatebenefratelli-Sacco, University of Milan, 20157 Milan, Italy; bernardo.dellosso@unimi.it; 5“Aldo Ravelli” Center for Neurotechnology and Brain Therapeutic, University of Milan, 20142 Milan, Italy; 6Department of Psychiatry and Behavioral Sciences, Stanford University, Stanford, CA 94305-5717, USA; 7IRCCS Orthopedic Institute Galeazzi, Tourette’s Syndrome and Movement Disorders Centre, 20161 Milan, Italy; roberta.galentino@gmail.com (R.G.); sarademichele1@gmail.com (S.D.M.); carlotta.zanaboni@libero.it (C.Z.D.); mauroportamilano@gmail.com (M.P.); 8Faculty of Medicine and Surgery, University Vita e Salute San Raffaele, 20132 Milan, Italy

**Keywords:** food-drug interactions, herb-drug interactions, pharmacologic processes, dietary supplements, nutraceutical, precision medicine

## Abstract

Preclinical and clinical studies suggest that many food molecules could interact with drug transporters and metabolizing enzymes through different mechanisms, which are predictive of what would be observed clinically. Given the recent incorporation of dietary modifications or supplements in traditional medicine, an increase in potential food-drug interactions has also appeared. The objective of this article is to review data regarding the influence of food on drug efficacy. Data from Google Scholar, PubMed, and Scopus databases was reviewed for publications on pharmaceutical, pharmacokinetic, and pharmacodynamic mechanisms. The following online resources were used to integrate functional and bioinformatic results: FooDB, Phenol-Explorer, Dr. Duke's Phytochemical and Ethnobotanical Databases, DrugBank, UniProt, and IUPHAR/BPS Guide to Pharmacology. A wide range of food compounds were shown to interact with proteins involved in drug pharmacokinetic/pharmacodynamic profiles, starting from drug oral bioavailability to enteric/hepatic transport and metabolism, blood transport, and systemic transport/metabolism. Knowledge of any food components that may interfere with drug efficacy is essential, and would provide a link for obtaining a holistic view for cancer, cardiovascular, musculoskeletal, or neurological therapies. However, preclinical interaction may be irrelevant to clinical interaction, and health professionals should be aware of the limitations if they intend to optimize the therapeutic effects of drugs.

## 1. Introduction

Oral drug administration is easy, cost-effective, and often preferred by patients, because they are able to take medicine without assistance or a dedicated delivery device, therefore assuming pharmacological therapy independently and at home. However, drug efficacy may be modified by meals consumed concomitantly, as food matrices and bioactive compounds might interfere at different phases of the pharmacokinetic process. Specifically, food-drug interactions in the absorption phase of orally administered drugs are investigated by studies of food-effect bioavailability (BA) and fed bioequivalence (BE) [1], and proper information about the correct way of drug consumption should always be conveyed to the patient. However, the recent and increasing use of traditional medicine along with complementary and alternative medicine (CAM) approaches, such as dietary modifications and dietary supplements, overcomes the concepts of fed or fasting drug administration, as patients are exposed to novel foods and supplements that contain various food bioactive compounds, such as nutraceuticals or phytochemicals [2]. For instance, food molecules may interfere with proteins involved in drug metabolism, which in turn cause variable and limited drug bioavailability [3,4,5], even though precise and comprehensive reviews of practical implications are mostly unseen.

Recent preclinical and clinical studies have suggested that many food molecules could interact with drug transporters and metabolizing enzymes through different mechanisms, which are predictive of what would be observed clinically. The lack of information could be the basis of some cases of pharmacological inefficacy [6]. Other effects of these overlooked food-drug interactions could be associated with unstable control of symptoms [7], or increased rates of side effects or occurrence of unexpected adverse events [8], specifically for patients suffering from chronic diseases that require the use of multiple drugs. Advertised effects of food supplements can attract misguided individuals who face a needless waste of money or unsafe interactions [9]. CAM approaches are often abused by patients who seek alternative therapies, possibly augmenting the risk of side effects and pharmacological opposition [10]. Given this usage trend and variability in individual pharmacological therapies and dietary habits, an increase in potential food-drug interactions has recently appeared. Historical recommendations for drug administration primarily concern fed or fasting conditions, alcohol, caffeine, grapefruit juice (*Citrus x paradisi* M.), and St. John's wort (*Hypericum perforatum* L.). This review aims at pointing out the most relevant data concerning food influence on drug efficacy, by classifying the mechanisms through which food bioactive compounds may interfere with drug pharmacokinetic/pharmacodynamic (PK/PD) profiles. Furthermore, a panorama of existing preclinical, clinical, and bioinformatic evidence is provided.

### Methodology of the Review

Data from Google Scholar, PubMed, and Scopus databases were reviewed, focusing on publications regarding pharmaceutical, pharmacokinetic, and pharmacodynamic mechanisms through which food components may interact with drug metabolism. The search considered in vitro and in vivo preclinical studies, clinical trials, and bioinformatic approaches. The presence of relevant food bioactive compounds in foods was integrated from the most authoritative online databases FooDB v.1.0 (http://foodb.ca/), Phenol-Explorer v.3.6 (http://phenol-explorer.eu/), and Dr. Duke’s Phytochemical and Ethnobotanical Databases v.1.9.12.6-Beta (https://phytochem.nal.usda.gov/). Targets, such as transporters, enzymes, and receptors were characterized for bioinformatic and functional information from DrugBank v.5.1.1 (https://www.drugbank.ca/), UniProt (https://www.uniprot.org/), and IUPHAR/BPS Guide to Pharmacology v.2018.4 (http://www.guidetopharmacology.org/). Unique bioactivity data of food molecules against drug target proteins was reviewed from the food-drug interactome map [11], managing to link the chemical space of diverse food molecules to drug target space. The cytochrome P450 drug interaction table of Indiana University School of Medicine (http://medicine.iupui.edu/) was used to integrate literature results relating to pharmacokinetic interactions.

## 2. Types of Food-Drug Interactions

Food-drug interactions can be broadly classified as occurring at (i) pharmaceutical (compatibility, solubility, stability), (ii) pharmacokinetic (absorption, distribution, metabolism, excretion), or (iii) pharmacodynamic (clinical effect) level, and are the result of physical or chemical exchanges between a food molecule and a drug. More precisely, another classification defined four types of food-drug interactions according to the distinction between pre-systemic and post-systemic phases [12]:Type I interactions (pharmaceutical), which refer to ex vivo bio-inactivations that usually occur in the delivery device with reactions of hydrolysis, oxidation, neutralization, precipitation, or complexation.Type II interactions, which affect the function of an enzyme (subtype A interactions, pharmacokinetic) or a transport mechanism (subtype B interactions, pharmacokinetic) before systemic circulation, in turn altering the absorption and bioavailability of oral or enteral administrations. Complexation, binding, or other deactivations may occur in the gastrointestinal tract (type C interactions, pharmaceutical).Type III interactions (pharmacokinetic), which occur after entrance into systemic circulation and involve changes in tissues distribution, penetration, or metabolism.Type IV interactions (pharmacokinetic), which denote affections in drug or food component clearance because of influences upon renal or enterohepatic excretion.

Although more accurate, this latter classification omits pharmacodynamic interactions, which could be considered the indirect result of previous pharmaceutical or pharmacokinetic interactions that could ultimately alter the clinical effect of the drug. However, recent direct pharmacodynamic interactions were discovered through bioinformatic approaches, as food molecules were shown to possibly act against same drug targets and compete for target-binding. Food-drug interactions happen continuously and daily, but they are considered relevant only if they predispose to treatment failure or adverse events, which are still hard to associate with food interference.

## 3. Pharmaceutical Interactions

Pharmaceutical interactions comprise types I and IIC food-drug interactions, and are due to the fact that the food matrix might directly react with medicines or indirectly alter their absorption by changing the gastrointestinal milieu [13]. These types of interactions have primarily been a matter of concern in intensive care units for many years, where the combination of multiple drugs is common, and therefore decisions of proper route, dose, and timing of drug administrations are always made considering food interferences. Knowledge of compatibility, solubility, and stability is necessary to avoid alterations upstream of the pharmacological response. When considering pharmaceutical interactions, drugs are classified according the Biopharmaceutics Classification System (BCS), which considers aqueous solubility and intestinal permeability [14]. When combined with the dissolution of the drug product, drugs are classified as follows: (i) class I (high solubility-high permeability), (ii) class II (low solubility-high permeability), (ii) class III (high solubility-low permeability), and (iv) class IV (low solubility-low permeability). Food matrix effects on drug bioavailability are least likely to occur with rapidly dissolving class I drugs, because their absorption is usually pH- and site-independent, and thus insensitive to differences in dissolution. However, food can indeed influence drug bioavailability when there is a high first-pass effect, extensive adsorption, complexation, or instability of the drug substance in the gastrointestinal tract, thus affecting the plasma maximum concentration of the drug (*C*_max_) and time to reach plasma maximum concentration (*T*_max_) by delaying gastric emptying and prolonging intestinal transit time. Concerning class II, III, and IV drug products, food, but also alcohol, is most likely to impact the in vivo dissolution or the absorption [1]. The BCS was reported to be integrated by the Biopharmaceutics Drug Disposition Classification System (BDDCS), which replaced the permeability criteria with the metabolism extent, with class I and II drugs being high permeable and class III and IV drugs being low permeable [13]. For instance, high permeability drugs [14], and also food components, are caffeine (coffee, energy drinks, sugary or sugar-free drinks) and theophylline (cocoa, tea). Type I interactions are far more common with drugs and nutrients administered intravenously, as they may be mixed in the delivery device and form precipitates for physical incompatibility. Conversely, type IIC interactions can be considered part of enteral or oral feeding practice, where chelation between divalent and trivalent cations with drugs may occur [12]. Also, polyvalent cations in milk and dairy products were reported to interact with some tetracyclines [15] and fluoroquinolones [16]. Indeed, it is important to consider the physiochemical milieu of the gastrointestinal tract, as food consumption delays gastric emptying, stimulates bile flow, changes gastrointestinal pH, increases splanchnic blood flow, changes luminal metabolism of a drug substance, and physically or chemically interacts with a dosage form or a drug substance [1]. For instance, chronic administration in rats of *Capsicum annuum* L. reduced acetylsalicylic acid, likely as a consequence of capsaicin (*Capsicum* genum) gastrointestinal effects [17].

Fed conditions state that the drug product is administered 30 min after start of the recommended meal, and with a glass of water. No food is allowed for at least 4 h post-dose. Water can be allowed as desired except for 1 h before and after drug administration [1]. Effect of meal per os can be considered a type IIC interaction, with an indirect influence on the drug rate and extent of absorption. Meals mainly affect drug absorption by way of *T*_max_, but do not necessarily lead to a reduction of the area under the concentration-curve (AUC) [12]. If food delays the onset of drug action, the efficacy might not be affected, despite differences in the pharmacokinetic profile. Conversely, the efficacy of some drugs, such as sildenafil, may be *C*_max_-driven and the delay of *T*_max_ due to the concomitant food consumption can affect clinical efficacy. In fact, when food intake causes a significant change in the AUC of a drug, the overall amount absorbed could be reduced and would cause a far more clinical effect. An exception appears to be levodopa and its interactions with dietary proteins: although a delayed absorption in the presence of food, with no changes in AUC as compared to without food, clinical efficacy was deeply affected [18]. The transit time of a drug product and its luminal dissolution, together with its permeability and systemic availability, are greatly affected when the administration occurs shortly after a meal is ingested. Dietary proteins were found to interact with the antiepileptic agent phenytoin, thus causing a reduced absorption of the drug [19]. Dietary fiber may act as a physical barrier and prevent drug contact with the gastrointestinal mucosa, but it can also directly adsorb drug molecules [13]. Soluble fibers, such as pectin, glucomannan, psyllium, and guar gum, which are able to form highly viscous solutions in water, were reported to interfere with the absorption of various medications, as well as insoluble components, such as cellulose or bran [20].

High-energy and high-fat meals are more likely to affect the gastrointestinal physiology and to facilitate intestinal uptake of the more lipophilic drugs [1], and therefore they are considered as test meals for food-effect studies. In general, high-fat meals increase the absorption of hydrophobic drugs through enhanced solubilization, but can conversely stimulate the formation of drug-bile micelles and decrease hydrophilic drug absorption [13]. The absorption of the antiretroviral azatanavir showed an increased AUC when administered with food [21]. For instance, a recommended meal for tests would comprise two eggs fried in butter, two strips of bacon, two slices of toast with butter, one hundred grams of hash brown potatoes, and about 220 grams of whole milk [1]. Food lipids promote the release of cholecystokinin, which slows gastrointestinal motility and increases the contact time between the drug molecules and the intestinal epithelial tissues. Indeed, decreased gastrointestinal motility increases the contact time between the drug and epithelial tissue, thus potentially allowing more complete absorption into the portal vein [12]. The content of fat in meals should be accounted for oral chemotherapeutics [22]. Conversely, significant decreased gastrointestinal tone may cause toxicities because of a more complete absorption, whereas decreased motility may augment the risk of clinical inefficacy [12]. Serotonin pathways in the gastrointestinal tract promote motility and foods rich in serotonin may shorten physical contact time [2]. Administration under fasting conditions refers to the consumption of the medicine with a glass of water after an overnight fast of at least 10 h. No food is allowed for at least 4 h post-dose, and water intake is allowed as desired except for 1 h before and after drug administration.

## 4. Pharmacokinetic Interactions

Pharmacokinetic interactions are types IIA, IIB, III, and IV. Absorption and excretion mechanisms, together with tissue metabolism, depend on transporters and metabolizing enzymes. Transepithelial and transendothelial fluxes comprise transporters across the intestine, the liver, the kidney, and brain capillaries. In general, transporters that exert a major role in xenobiotic metabolism include two major superfamilies: the solute carrier (SLC) superfamily, which comprises forty-eight subclasses, and the ATP-binding cassette (ABC) superfamily, which includes seven subclasses. The SLC superfamily encompasses a variety of transporters, such as the organic anion transporters (OAT) and the organic cation transporters (OCT) of family 22. Formerly known as SLC family 21, the SLCO family is now grouped as the organic anion transporting polypeptides (OATPs). ABC proteins are primary ATP-dependent and comprise seven subclasses (from ABCA to ABCG). They are known to be involved in the excretion of different molecules, thus decreasing the bioavailability of certain compounds. The purpose of these efflux transporters is believed to minimize host xenobiotic exposure, prevent the saturation of cytoplasmic enzymes, or provide repeated opportunities to metabolize the xenobiotic compound [12]. Active efflux proteins include the multidrug resistance protein 1 (MDR1), also known as P-glycoprotein 1 (P-gp), of family ABCB1, the canalicular multidrug resistance-associated protein 1 and 2 (MRP1 and MRP2) of family ABCC1 and ABCC2, respectively, and the breast cancer resistance protein (BCRP) of family ABCG2. MDR1 seems to manage preferably amphipathic cationic compounds and is representative among efflux transporters, thus tightly regulating how quickly and how much drug can reach the metabolizing enzymes located in the endoplasmic reticulum.

Xenobiotic metabolism is primarily accounted for by the highly polymorphic and inter-individual variable cytochrome P450 enzyme families (CYP450), whose activities are majorly concentrated in the liver and in the enteric villous tips from the duodenum to the distal jejunum, with a minor role for extrahepatic enzymes located in the lungs and brain. They are an extensive family of hem-containing monooxygenases with a huge range of both endogenous and exogenous substrates, and primarily mediate phase I reactions, such as oxidation, reduction, and hydrolysis. Single CYP has the capacity to metabolize structurally diverse molecules, and a single substrate can be metabolized by different CYPs. Additionally, CYPs exert their catalytic activity at different regions on the molecule, giving the promiscuous substrate binding sites in the enzyme. In humans, three families of CYPs mostly involved in xenobiotic metabolism are the CYP1, CYP2, and CYP3. CYP3A4 is representative among these enzymes, as it metabolizes a vast range of drugs in the liver, such as benzodiazepines, antidepressants, chemotherapeutic agents, and calcium channel blockers. Phase I reactions are also mediated by flavin-containing monooxygenases (FMO). Other oxidative reactions are catalyzed by alcohol dehydrogenases (ADH) and monoamine oxidases (MAO), respectively localized in the cytosol and on the mitochondrial outer membrane of the liver. Phase II reactions involve other metabolizing or conjugating enzymes, such as sulfotransferases (SULT), UDP-glucuronosyltransferases (UGT), NAD(P)H:quinone oxidoreductase (DT-diaphorase), epoxide hydrolases (EPH), glutathione S-transferases (GST) and N-acetyltransferases (NAT), all of which produce end products more readily excreted. The conjugation reactions, by promoting excretion in the bile and/or the urine, affect detoxification.

For better comprehension, pharmacokinetic interactions can be divided into pre-systemic and post-systemic food-drug interactions. The pre-systemic involves the first-pass metabolism of xenobiotics, which takes place first in the intestine and then in the liver after transit in the bloodstream, with the stomach accounting for a minor role. Post-systemic concerns food-drug interactions that occur after entrance into the systemic circulation, comprising tissue penetration and metabolism. Eventually, proteins targeted by food bioactive compounds at the pre-systemic and post-systemic levels may overlap, as they can be expressed in multiple cells.

### 4.1. Pre-Systemic Interactions

Absorption mechanisms across intestinal epithelial cells is vectoral in nature, as cells are polarized and organized in monolayers. OATPs and SLC transporters primarily carry out both gut and hepatic absorption of amphipathic organic anions and hydrophilic organic cations, respectively, accounting for the uptake of many food components and drugs. Enteric ABC proteins excrete xenobiotics back into the gut lumen. Drug resistance may occur because of a decreased uptake or an enhanced efflux of drugs, which is associated to hydrophobic anticancer refractoriness because of the overexpression of these transporters in tumor cells. On the apical side of intestinal epithelia, efflux proteins include the MDR1, the MRP2, and the BCRP. MRP3 of family ABCC3 is expressed on the basolateral membrane of epithelial cells. ABCs are also located on the blood-facing membrane of hepatocytes and on the bile canalicular membrane where they export compounds into the bile. Hepatic drug uptake, followed by metabolism and excretion, is the major determinant of the pre-systemic clearance since liver activity ultimately determines systemic blood levels. Nevertheless, the inhibition of these influx and efflux transporters by bioactive food compounds may alter the synchronization of the enteric/hepatic transport-metabolic systems, thus leading to a significant change in a drug oral bioavailability and a relevant food-drug interaction. Of note, an atypical food-drug interaction was reported [23], with black pepper, red pepper, and ginger increasing microvilli length and perimeter in animals, thus potentially facilitating the overall absorption capacity of the small intestine.

#### 4.1.1. Food Interference with Enteric and Hepatic Transport

Grapefruit is a classic example of both drug transportation and CYP450 drug oxidation modulator, as it was stated to exert an opposite effect on xenobiotic transporters and metabolizing enzymes. It was reported to inhibit drug absorption catalyzed by the intestinal OATP1A2, thus lowering plasma concentration of substrates [24]. In particular, naringin of grapefruit and hesperidin of *Citrus* spp. proved in vitro to inhibit human OATP1A2, and naringin showed to be clinically relevant [25]. It has also been reported that the major constituents of grapefruit, as well as orange (*Citrus x sinensis* L.) juice, significantly inhibited the OATP2B1 function in vitro [26]. Concomitant administration of Seville orange (*Citrus x aurantium* L.) or apple juice with fexofenadine decreased drug absorption in rats, likely acting on OATPs [27]. In fact, significant inhibition of OATP drug transporters was observed not only for grapefruit and orange juice, but also for apple juice [28]. In particular, apple showed to inhibit non-human OATP2B1 in vitro [29], easily attributable to the similar composition of flavonoids between grapefruit, orange, and apple. In a clinical trial, grapefruit juice at a commonly consumed volume diminished oral bioavailability of a drug substrate of the intestinal OATP1A2, likely acting as inhibitor [30]. A reduction of fexofenadine AUC and *C*_max_ was observed in humans after consumption of orange and apple juice [31]. Apigenin (marjoram—*Origanum majorana* L., parsley—*Petroselinum crispum* L., Chinese cabbage, ginkgo—*Ginkgo biloba* L.), kaempferol (capers, cumin—*Cuminum cyminum* L., caraway—*Carum carvi* L., dill—*Anethum graveolens* L., common pea, tarragon—*Artemisia dracunculus* L., cabbage—*Brassica oleracea var.* Capitate L., garden cress—*Lepidium sativum* L., ginkgo), and quercetin (onion, Chinese cabbage, capers, black elderberry, ginkgo) competitively inhibited enteric OATP1A2 and OATP2B1 in vitro [32]. The hepatic transporters OATP1B3 and OATP2B1, which are involved in drug uptake on the basolateral membrane, were inhibited in vitro by the various flavonoids from ginkgo (apigenin, kaempferol, quercetin) and from grapefruit (naringenin, naringin, rutin—quercetin glycoside). No effect on the hepatic OCT1 was found [33]. Bioinformatic evidences confirmed the bioactivity of naringenin, naringin, and quercetin against OATP1B1, OATP1B3, and OATP2B1 [11]. Same transporters can be targeted by daidzein (soy bean, soy products), genistein (lima bean—*Phaseolus lunatus* L., soy bean and products), and glycyrrhizin (liquorice) [11]. Moreover, caffeine acts not only against OCT1, but also OATP1B1, OATP1B3, and OATP2B1 [11]. Data is summarized in Table 1.

MDR1 is expressed in tissues that are exposed to xenobiotics, such as the apical surface of jejunal and colonic cells, but also in the canalicular surface of hepatocytes, with both MDR1 and MRP2 governing active biliary excretion [34]. Grapefruit and orange juices inhibited in vitro the intestinal MDR1 and MRP2 [35], potentially counterbalancing the action on OATPs. Intestinal MDR1 activity was inhibited in another in vitro study by extracts of bitter melon, soybean, dokudami and Welsh onion, with authors attributing to 1-monopalmitin of bitter melon the most potent inhibition [36]. Diosmin increased the apical-to-basal transport and decreased the efflux transport in human intestinal cells in vitro [37], thus supporting the inhibitory action on MDR1 by Citrus spp. Methoxyflavones of orange and tangerine (*Citrus reticulata* B.), such as tangeretin, were suggested to be responsible for the specific inhibition of intestinal MDR1 in vitro [38]. Bergamottin, tangeretin, and nobiletin were shown to potently inhibit BCRP in a concentration-dependent manner [39], and. in particular. nobiletin showed an inhibitory effect on MDR1 and MRP1 [40]. Sulforaphane from cruciferous vegetables, such as broccoli (*Brassica oleracea* var. Italica), cauliflower (*Brassica oleracea* var. Botrytis), radish (*Raphanus raphanistrum* subsp. *sativus*), but also watercress (*Nasturtium officinale* W.T.A.), is an isothiocyanate forming from the hydrolysis of glucosinolates by the action of the co-existing, but physically segregated, enzyme myrosinase. Sulforaphane increased intestinal protein levels of MRP1 and MRP2 in vitro [41]. Long-term feeding with ginseng (*Panax* L.) induced the expression of intestinal MDR1 expression in rats, thus decreasing the bioavailability of fexofenadine [42]. Sesamin of sesame, ginkgolic acid of ginkgo, and glycyrrhetinic acid and glabridin of liquorice, inhibited MDR1 of cells in vitro, with glycyrrhetinic acid inhibiting also the function of MRP1 [43]. In a clinical trial, repeated ingestion, but not single oral dose, of ginkgo was able to increase *C*_max_ and AUC of talinolol, likely due to the inhibition mechanism of intestinal MDR1 and MRP2 that overwhelmed the inhibition of intestinal OATPs [44]. Piperine of *Piper* genus inhibited human MDR1 [45]. Rosemary and common sage (*Salvia officinalis* L.) phytochemicals, such as carnosic acid, showed inhibitory effects on MDR1 [46]. Soymilk and miso significantly induced MDR1 in an animal study [47]. Bioinformatic evidences showed that resveratrol (red currant, red wine) could act against MDR1, and that apigenin, biochanin A (peanut, soy bean), genistein, glycyrrhizin, naringenin, quercetin, and resveratrol act against MRP2 [11]. Data is summarized in Table 2.

#### 4.1.2. Food Interference with Enteric and Hepatic Metabolism

Grapefruit is known as an inhibitor of both intestinal CYP3A4 and CYP1A2, thus causing an increased *C*_max_ and AUC [24], with more risk for side effects/adverse outcomes. Grapefruit juice also proved to have an in vitro inhibitory potential on hepatic microsomal subfamily CYP3A [48], whose gene locus includes A4, A5, A7, and A43 members. This food-drug interaction was ascribed to different grapefruit constituents, such as furanocoumarins (bergamottin, bergapten), and flavonoids (e.g., naringin, naringenin, quercetin, and kaempferol) [49]. Bioinformatic studies confirmed that kaempferol, bergapten, naringenin, quercetin, tangeretin, and taxifolin (sweet orange, Mexican oregano) could interact with CYP3A4 [11]. In humans, quercetin inhibited CYP1A2 function, but enhanced CYP2A6 and NAT activity [50]. Ethanol is known to induce CYP2E1 [51]. Casticin (chasteberry—*Vitex agnus-castus* L.) and resveratrol were shown to interfere in vitro with both CYP3A4 and CYP2C9 [52]. Resveratrol activity against CYP3A4 was confirmed by bioinformatic approaches [11]. Resveratrol was non-competitive reversible inhibitor of CYP2E1 in rat and human liver cells [53]. Cranberry (*Vaccinium macrocarpon* A.) was involved in the possible potentiation of warfarin anticoagulation, as its content of anthocyanins, flavonols, proanthocyanidins, and phenolic acid derivatives was associated with the in vitro inhibition of CYP3A and CYP2C9 [54], but relevant pharmacokinetic interactions were excluded [55]. Conversely, significant clinical effects were caused by grapefruit and Seville orange juices [28]. Same food components of grapefruit juice are present in other foods, such as pomegranate (*Punica granatum* L.) and black mulberry (*Morus nigra* L.), thus accounting for some time- and concentration-dependent inactivation of hepatic CYP3A [48]. Conversely, an animal study suggested that both pomegranate and grapefruit components could act primarily on enteric, but not hepatic, subfamily CYP3A [56]. Intestinal effects of pomegranate juice were also confirmed by the increase AUC of tolbutamine in rats [57]. Epicatechin gallate (green tea, peppermint—*Mentha × piperita* L., common grape—*Vitis vinifera* L.) and tangeretin showed an in vitro potent inhibition of subfamily CYP1A in human liver microsomes [58], but tangeretin failed to have appreciable clinical effects [59]. Epicatechin gallate was reported to have bioactivity also against CYP3A4 [11]. Apple juice extract inhibited the catalytic activity of CYP1A1[60]. Sulforaphane induced the phase I bioactivating enzyme CYP1A1 [61], but reduced the expression of CYP3A4 [62]. The enzymes CYP1A2, CYP3A, and CYP2C11 are involved in caffeine metabolism [63], and theophylline acted on CYP1A2 as substrate inhibitor [51]. Theobromine (cocoa powder, coffee) may interfere with CYP3A4 [11]. In rats, diallyl disulphide from soft-necked garlic (*Allium sativum* L.) and piperine induced CYP1A1, CYP1A2, CYP2B1, CYP2B2. CYP2E1 was inhibited by diallyl disulphide and induced by piperine. CYP3A4 was agonized by diallyl disulphide and by glycyrrhizin [64], but inhibited by piperine [45]. Diallyl disulphide was confirmed to be substrate of CYP3A4 [11]. Liquorice also induced CYP1A2 in vivo [65], and most members of CYPs [4]. Tomatoes (*Solanum lycopersicum* L.) and tomato-based products, such as ketchup, contain numerous phytochemicals, of whom lycopene is the best known. In rats, lycopene supplementation increased hepatic CYP2E1 [66]. Capsaicin, and ellagic acid (chestnut, blackberry, black raspberry) were reported to strongly inhibit in vitro CYP2A2, 3A1, 2C11, 2B1, 2B2 and 2C6 [67], with possible induction of CYP1A2 [68]. Capsaicin may be also substrate of CYP3A4 [11]. Hesperetin (lime, sweet orange) selectively inhibited human CYP1A1 and CYP1B1 [69], and is substrate also of CYP3A4 [11]. Soymilk and miso significantly induced CYP3A4 in an animal study [47]. Other than the abovementioned food components, bioinformatic studies showed different molecules to potentially interact with CYP3A4: 6-shogaol (ginger—*Zingiber officinale* R.), apigenin, biochanin A, caffeic acid (common verbena—*Verbena officinalis* L., tarragon, sweet basil, common thyme—*Thymus vulgaris* L.), curcumin, daidzein, diosgenin (fenugreek—*Trigonella foenum-graecum* L., carrot), estragole (anise—*Pimpinella anisum* L., fennel—*Foeniculum vulgare* M., sweet basil, tarragon), galangin (Mexican oregano), genistein, geraniol (common grape, black walnut—*Juglans nigra* L., common thyme, carrot), harmine, luteolin (Mexican oregano, globe artichoke—*Cynara cardunculus* var. Scolymus L., thyme, sage, anise, angelica—*Angelica keiskei* I., broccoli), myricetin (common walnut, fennel, European cranberry, blackcurrant, black crowberry, Romaine lettuce—*Lactuca sativa* var. Longifolia L., highbush blueberry, strawberry), pyrocatechol (Arabica coffee), rutoside, safrole (nutmeg, sweet basil, rosemary), sesamin, and xanthotoxin (parsnip, parsley) [11]. In a clinical trial, charcoal-broiled meat induced CYP1A enzymes in the liver and small intestine of healthy volunteers [70]. Decreased CYP1A2 activity and increased CYP2A6 activity was displayed in women by genistein [71]. Genistein may be also substrate of CYP3A4 [11]. Watercress inhibited subfamily CYP2E1 and increased AUC of a substrate of the CYP in a clinical trial [72]. In a clinical trial, CYP1A2 activity was concluded to be increased by an enriched-brassica diet, but decreased by an enriched-apiaceous diet (dill, celery, parsley, parsnip—*Pastinaca sativa* L.) [73]. The parsnip constituent imperatorin and the parsley component isopimpinellin may have activity against CYP3A4 [11]. Resveratrol was shown to increase *C*_max_, AUC, half-life, and decrease the elimination rate constant of substrates of CYP2C9 [74] and CYP2E1 [75] in healthy volunteers through an inhibitory mechanism. Bioinformatic approaches reported that CYP1A2 is targeted by apigenin, galangin, hesperetin, isorhamnetin (green bell pepper, dill, fennel, chives, tarragon), kaempferol, luteolin, naringenin, pinocembrin, and quercetin [11]. Furthermore, bergapten, furanocoumarin, and xanthotoxin can act against CYP2A6, 2B6. Moreover, apigenin, bergapten, biochanin A, capsaicin, curcumin, daidzein, dihydrocapsaicin, ellagic acid, furocoumarin (fig, parsley), galangin, genistein, hesperetin, kaempferol, luteolin, myricetin, naringenin, nicotine, pinocembrin, quercetin, resveratrol, sesamin, and xanthotoxin are substrates of CYP2C19 [11]. Apigenin, bergapten, biochanin A, caffeic acid, capsaicin, curcumin, daidzein, dihydrocapsaicin, ellagic acid, furocoumarin, galangin, genistein, hesperetin, kaempferol, luteolin, myricetin, naringenin, quercetin, resveratrol, and sesamin have activity against CYP2C9 [11]. The following phytochemicals were reported to have activity against CYP2D6: 6-shogaol, apigenin (marjoram, parsley, Chinese cabbage), bergapten (anise, fig, parsnip, parsley, celery stalks, grapefruit), β-sitosterol (cherimoya—*Annona cherimola* L., canola, sesame, pistachio, common buckwheat), capsaicin, curcumin (turmeric, curry powder), daidzein, dihydrocapsaicin, formononetin (soy bean), galangin, harmine, hesperetin, kaempferol, luteolin, myricetin, pinocembrin (Mexican oregano), quercetin, resveratrol, phytoserotonin, sesamin (sesame seeds), and xanthotoxin [11]. Data is summarized in Table 3.

Concerning phase II enzymes, an in vitro study showed DT-diaphorase activity to be increased by watercress, garden cress, and mustard (*Sinapsis alba* L.), whereas UGT activity was shown to be augmented by garden cress and mustard, with a further agonist activity on NAT by mustard [76]. In humans, NAT activity was also enhanced by quercetin [50]. Milk thistle (*Silybum marianum* L.), the green tea catechin epicatechin gallate, saw palmetto (*Serenoa repens* JKS), cranberry, quercetin, and kaempferol inhibited in vitro different members of the UGT family of enzymes in liver microsomes [77]. Bioinformatic studies reported (+)-Catechin, apigenin, ergosterol (common wheat), galangin, menthol, naringenin, carvacrol (pot marjoram, common thyme, black walnut, winter savory), diosgenin, eugenol (cloves, allspice—*Pimenta dioica* L., carrot), linalool (spearmint, coriander, common thyme, winter savory, cardamom, sweet basil, sweet bay—*Laurus nobilis* L., orange mint—*Mentha citrate* E.), naringin (rosemary, grapefruit/pomelo hybrid, grapefruit, pomelo—*Citrus maxima* M), quercetin, tea polyphenol, borneol (common sage, cardamom, rosemary, pot marjoram, common thyme, spearmint, winter savory), citronellol (ginger, sweet basil, winter savory), ethinyl estradiol (cloves, cardamom, ginkgo, nuts, nutmeg, star anise, ginger, Ceylon cinnamon), shikimic acid (mammee apple—*Mammea americana* L.), and caproic acid (milkfat, butter, cheese, cream, coconut) to be substrates of different UGTs [11]. The polyphenol punicalagin from pomegranate was able to inhibit enteric SULT, likely further affecting the availability of drugs [78]. Sulforaphane induced the activity of phase II drug detoxificating enzymes DT-diaphorase, GST [79], and SULT [80]. Of note, by comparing gene expression signatures of different foods and drugs, broccoli was shown to have the highest number of connections with drugs [11]. The induction of hepatic SULT was also reported for asparagus, eggplant, cauliflower, celery, carrot, and potato [80]. EPH and GST were increased in mice by freeze-dried soya bean, Brussels sprout, cauliflower, and onion [81]. Data is summarized in Table 4.

### 4.2. Post-Systemic Interactions

Food-drug interactions that happen after the entry into systemic circulation are part of post-systemic interactions, comprising plasma protein interactions, and have rather similar interferences with proteins or metabolizing enzymes for tissue penetration or metabolism, respectively. Of note, both pre- and post-systemic metabolisms are also indirectly influenced by blood delivery through the hepatic portal vein, which is increased by the splanchnic blood flow associated to the presence of food.

#### 4.2.1. Food Interference with Blood Transport

Drugs may freely circulate in blood or can be bound to serum proteins, and possible food-drug interactions occur at the level of blood transport. These types of interactions were mostly studied for anticonvulsants, which can be displaced from albumin-binding by fat emulsions of co-administered parenteral nutrition formula [82]. Serum albumin is known to bind a large number of molecules, including caffeine. In particular, polyphenols were shown to interfere with drug pharmacokinetics by acting on albumin-binding, which may ultimately result in increased drug uptake and/or elimination. In vitro displacement of albumin site II (in subdomain IIIA of naproxen) was shown for resveratrol and casticin, but no interference with site I (in subdomain IIA of warfarin-azapoprazone) was shown [52]. Salvianolic acid B and rosmarinic acid (*Salvia miltiorrhiza* B.) were reported to bind to bovine serum albumin through interaction with subdomain IIA [83], with salvianolic acid B being able to deeply interfere with warfarin subdomain in the human counterpart [84]. However, even if polyphenols were shown to form stable complexes with serum albumin, very small structural conformational changes may occur, thus not significantly displacing drugs. Moreover, different albumin binding sites and affinities for food components [85] may increase the uncertainty for relevant food-drug interactions. A partial albumin unfolding was reported upon binding with curcumin or genistein [86], but no significant perturbation for quercetin [87] binding.

#### 4.2.2. Food Interference with Renal and Brain Transport, and Post-Systemic Metabolism

OCTs contribute to the permeability of the blood–brain barrier, whereas the members of organic anion transporting polypeptide (OATP1A4 and OATP1A5) and organic anion transporter (OAT3) families are involved in the efflux of organic anions from the CNS. OATs are mainly responsible for renal tubular secretion from blood to urine, such as OAT1 and OAT3 on the basolateral side of the membrane, but also renal reabsorption from urine to blood, such as OAT4 on the apical side of the membrane [88]. ABC transporters are highly expressed in the endothelial cell surface of the blood brain barrier (BBB), choroid plexus epithelial cells [89], and renal proximal tubule [34]. The bioavailability for some acidic or basic drugs that rely on a tubular passive permeation mechanism is affected by changes of luminal pH. Dietary proteins/protein food tend to make urine more acidic [90], thus rendering weakly acidic drugs better reabsorbed. Conversely, if urine is alkaline, then weak bases tend to be more easily reabsorbed. Components of licorice were shown to be substrates of OAT1 [91]. The inhibition of efflux mechanisms of MDR1, MRP4, or BCRP in the BBB would cause an increase in the concentration of substrates in the CNS, as was observed for MDR1 and antipsychotics [92]. Conversely, the induction of ABC transporter expression would decrease central bioavailability of substrates, as was reported for MDR1 and fexofenadine in rats fed long-term with ginseng [42]. ABC secretory transporters in kidneys mainly comprise MDR1 for organic cations and MRP2 or MRP4 for organic anions. In rat renal proximal tubules, isothiocyanates inhibited BCRP and caused an increase of AUC of the substrate in rat renal proximal tubules [93]. Lycopene was shown to inhibit in vitro post-systemic bio-activating CYP1A1 and CYP1B1, with a concomitant increased microsomal detoxifying UGT activity [94]. In conclusion, dysfunction of both P-glycoprotein and BCRP exhibits synergy on the increase in the brain-to-plasma concentration ratio of the common substrates. Evidence of post-systemic pharmacokinetic interactions came from bioinformatic studies: apigenin, quercetin, naringenin, resveratrol and nicotinic acid were found to have the strongest binding affinities with drug ADME proteins [11].

## 5. Pharmacodynamic Interactions

A variety of pharmacodynamic interactions has been observed in preclinical, clinical, but also bioinformatic studies, which consider the predicted bioactivity profile, as the chemical space of food components was linked with drug target space. Foods that contained the most interfering components with drug targets, such as receptors, were found to be ginger, camellia tea, poppy seed, beansprout, strawberry, tomato, swede, fennel, celery, licorice, guava, sugar-pea, mango, maize, turnip, rice, and avocado. Apparently, food components showed a main bioactivity towards proteins involved in signal transduction [11]. Specifically, the food-drug interactome map underscored several pharmacodynamic food-drug interactions. Naringenin can target aromatase, and kaempferol has a high affinity for the epidermal growth factor receptor. Phytoserotonin (sea buckthorns—*Hippophae rhamnoides* L., bananas, chicory, Chinese cabbage, coffee powders, green coffee bean, green onion, hazelnut, kiwi, lettuce, nettle, paprika, passion fruit, pawpaw, pepper, pineapple, plantain, plum, pomegranate, potato, spinach, strawberry, tomato, velvet bean, wild rice, sunflower, potato, and tomato) [2], interacts strongly with 5-hydroxytryptamine receptors. P-hydroxybenzoic acid (coconut, currant, sprouted lentil, swede) binds strongly to carbonic anhydrase. Resveratrol was found to interfere with the estrogen receptor and prostaglandin G/H synthase 2, but also caffeic acid, daidzein, genistein, kaempferol, naringenin, quercetin, rutoside can act against estrogen receptors [11]. Acetylcholinesterase and cholinesterase are targets of quercetin, apigenin, galangin, kaempferol, luteolin, myricetin, rutoside, harmine, hesperetin, kaempferol, luteolin, myricetin, naringenin, rutoside, eriodictyol (Mexican oregano, lemon, marjoram, lemon balm—*Melissa officinalis* L.) [11]. Potential interactions with glutamate receptors were reported as glutamate/glutamic acid was described to be present in caviar, cheese, crackling, chips, dried cod, fermented beans, fish sauces, gravies, instant coffee powder, meats, miso, mushrooms, noodle dishes, oyster sauce, Parmesan cheese, ready-to-eat meals, salami, savory snacks, seafood, seaweeds, soups, soy sauces, spinach, stews, tomato, and tomato sauce [2]. Higenamine (custard apple) was reported to act on D(2) dopamine receptor and D(1A) dopamine receptor was found to be targeted by taxifolin, alpha-aminoadipic acid (rye, wheat, broccoli, dill, chard), dihydrocapsaicin, myricetin, quercetin, and phytodopamine (chard, oat, red beetroot, but also aubergine, avocado, banana, common bean, apple, orange, pea, plantain, spinach, tomato, and velvet bean) [2]. β_2_ adrenergic receptor is targeted by liquiritin [11].

## 6. Discussion

For many years, CAM approaches have been introduced to improve human health, and their role has evolved from folk medicine to worldwide usage, according to different attitudes and regulatory aspects. However, a dark side of CAMs exists [95,96]. A diverse range of food compounds constantly interact with proteins involved in drug PK/PD profiles, starting from the absorption phase to possible alterations in clinical efficacy, with potential involvement in pharmacological refractoriness, side effects of drugs or unexpected adverse events. Although the amount of food bioactive compounds that needs to be consumed in order to interfere with a specific mechanism can be far from the usual exposure, the potential for additive interference of more than one molecule or during chronic intake should be considered. Bioinformatic approaches can account for that aspect [11]. For instance, authors interviewing a large cohort of neuropsychiatric patients reported that about 80% of subjects using CAMs initiated the use without informing their doctor [97]. Patients consider their use of CAMs not worthy of note, or they feel uncomfortable in admitting turning to non-traditional medicine remedies. Usage of CAMs may be hidden during first anamneses with doctors, being subsequently exposed only during follow-ups [9]. Hospitalized patients may encounter a higher risk of food-drug interactions during hospital standard of care, especially if not investigated [98]. Subjects with impaired hepatic, renal, or gastro–intestinal function may be more vulnerable, together with patients taking several medications for chronic conditions. More caution should be considered for food components that inhibit metabolizing enzymes or interfere with the metabolism of hydrophobic drugs, whose accumulation in adipose tissue and cellular phospholipid bilayers can be increased. If the therapeutic window is narrow, then food interference may easily cause a relevant clinical effect [13]. Conversely, dietary recommendations do not exactly satisfy individual needs and CAMs, such as dietary supplements, can fill the nutritional gap between needs and intakes, thus ameliorating the nutritional status or avoiding specific heath issues [99]. In parallel, food was enriched or fortified in order to avoid either natural impoverishment [100] or the impact of cooking methods on nutritional values [101]. CAM approaches can attain confident results if well managed, for instance protein-redistribution diet in Parkinson’s disease treated with levodopa [102], low day-to-day variability in dietary vitamin K during warfarin therapy [7], low-fat meals for oral chemotherapeutics [22], regular water and sodium intake with lithium [103], or low-tyramine diet with monoamine oxidase inhibitors [104].

Both positive and negative abovementioned results in clinical practice are due to the fact that foods and beverages affect the way medicines exert their effects through diverse mechanisms, which can be classified as pharmaceutical, pharmacokinetic, and pharmacodynamic food-drug interactions. This is tuned to give the broadest view to the readers by providing a comprehensive classification of the major site of food-drug interactions, as pharmaceutical interferences, blood protein disruptions, and direct pharmacodynamic interferences are usually disregarded [3,4,5,13]. Furthermore, for the first time, evidence from bioinformatic approaches have been integrated with the literature data. Results are summarized in Table 5. Most of the data concerning food interference with drug metabolism comes from in vitro or animal studies, but preclinical models cannot be uniquely relied upon for predicting relevant human food-drug interactions, as large differences among species in the ability to metabolize xenobiotics exist. Moreover, in vitro evidence may be difficult to observe in humans, because of insufficient concentrations of causative food components at the drug active site. Pharmaceutical interactions for oral drugs may be avoided when preferring fasting. If a fed condition is required, health professionals should pay attention to the meal content of minerals, fiber, and macromolecules, such as proteins and lipids. Sometimes, a pharmaceutical interaction with food is needed in order reduce drug side effects [13]. Concomitant alcoholic beverage intake (wine, beer, spirits) is always forbidden, especially for modified-release systems that could encounter a rapid release known as alcohol-induced dose dumping [105].

Under in vitro conditions, a high amount of food bioactive compound may target cellular transporters or metabolizing enzymes, while varying levels occur in animals or humans because food molecules are extensively metabolized. Therefore, a relevant preclinical interaction may be a non-relevant clinical interaction. For example, few clinical effects were suggested when considering the interference of food components with plasma protein binding [106]. Food bioactive compounds bioavailability may be influenced by many factors, such as the luminal pH of the intestine [107], inter-individual variabilities, hydrophobic properties of the food molecule [108], or even drugs themselves. In fact, drug metabolism can directly alter the same steps of food components kinetic profile, and some medications may cause gastrointestinal dismotility or diarrhea, thus indirectly affecting food component bioavailability. The compositions of commercial juices varies according to brand, lot, and, in the case of homemade juices, method of preparation [109]. Additionally, different parts of the same plant, such as the stem or fruits, may have very diverse compositions also depending on subspecies and variety of plants, cultivar and ecotype, chemotype, soil and nourishment, geographical location, environmental impact during plant growth, seasons of growth and harvest, weather and climate changes, and agricultural practices [2]. Clinical trials showed significant pharmacokinetic interactions, whose practical consequences on pharmacological response were shown to vary. Of note, only grapefruit juice appeared to give the most promising clinical data, probably due to its now-dated interest. However, fruits from the *Citrus* genus, such as Seville orange or pomelos, may have the same effect as grapefruit juice and their consumption should be avoided if taking medicines that interact with grapefruit juice. Still, to study other suggested food-drug interactions in human clinical trials would be preferred in order to prevent their interactions before they are observed in daily practice.

A complete patient education should focus on the possible presence of the warning “do not crush”: if ignored by the patient, pharmaceutical interactions with food would be more likely to happen for special formulations, such as those that are enteric-coated or slow-release in nature. It should be impressed on patients not to swallow drugs with alcohol, drink alcohol after drug ingestion, or take medicine after alcohol consumption [110]. However, concomitant abstention from alcoholic beverages during drug administration should be further extended to chronic abstention, as an additive action on diverse pharmacokinetics aspects was noted, other than the well-known alcohol-induced dose dumping phenomenon. Indeed, for chronically ill subjects it would be hard to totally exclude alcohol from dietary habits because light alcoholic beverages, such as red wines, are considered part of the much-advertised Mediterranean diet. If a bioactive food compound acted uniquely on metabolizing enzymes, drugs that are a substrate of the same CYP would be poorly metabolized and would remain in the body for longer periods of time. Induction of CYPs that carry out the oxidation of substrates with the production of activated oxygen, for instance the case of the uncoupled CYP3A4 reaction, could lead to the rise of oxidative stress. Lack of relevant clinical evidence may be related to the short phase of food exposure, usually a few weeks for trials [55]. In fact, chronic and not-occasional consumption of a specific food may cause a relevant food-drug interaction, as interference with target proteins from a single exposure may last only a few days [56]. Evidence from animal studies has confirmed the minor affection of single dose exposure of a food bioactive compound, compared to repeated administrations [111]. Other than red wine, bioinformatic studies showed that most of the food molecules that bind to human proteins are also very common in the Mediterranean diet and in turn difficult to avoid. However, the main limitation of bioinformatic approaches is that even if a food component was found to be active against a drug target, it is not known if this interaction would cause an interference with the drug-binding mechanism [11]. What should be pointed out, however, is that the bioinformatic approach was able to identify sub-networks of foods that contain molecules interacting with specific proteins, possibly accounting for these aspects when medicines targeting the same proteins are prescribed.

## 7. Conclusions

To investigate individual variance, during the past few years research has turned to pharmacogenomics, thus pursuing the much-desired tailor-made therapy. However, even before reaching these levels of perfection, it is necessary to take a step backwards, as the optimization of drug compliance and the precise avoidance of food-drug interactions has not yet been reached. Knowledge of any nutraceutical or phytochemical residue, additive, contaminant, adulteration with non-declared ingredient, and dietary neurotransmitter is the condition *sine qua non* there is no well aware nutrition, and would provide a missing link for obtaining a holistic view in clinical practice. Then, it is important to have good knowledge about the type and number of binding sites that a certain food component will interact with on a given protein because this information would allow health professionals to estimate the degree of interference. Because of the opportunity to positively exploit interferences between foods and drug mechanisms, health professionals should be aware of relevant interactions in order to optimize the therapeutic effects of both prescribed and over-the-counter drugs. Figure 1 summarizes the major sites of food-drug interactions. For example, to prevent multidrug resistance, inhibitors of efflux transporters are considered a useful therapeutic approach, and food components may be considered more promising candidates for sensitizing patients to a particular drug [112]. It would be necessary to promptly make a change in a patient’s diet in order to increase therapeutic efficacy according to the disease category, such as for cancer and cardiovascular, musculoskeletal, or neurological diseases. Information about food-drug interactions should always be included in patient and caregiver education, starting with historical indications to avoid concomitant intake of alcohol or caffeine [9], but also integrating new recommendations about optimal timing, meal composition, and intake or avoidance of food bioactive compounds.

## Figures and Tables

**Figure 1 pharmaceutics-10-00277-f001:**
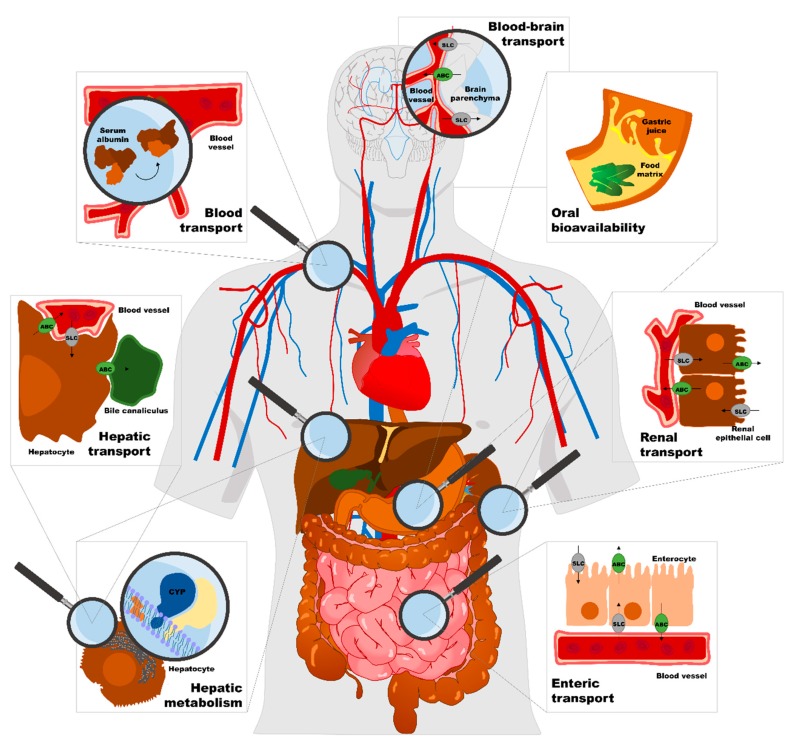
Major sites of food-drug interactions. SLC = solute carrier proteins, ABC = ATP-binding cassette proteins, CYP = cytochrome P450.

**Table 1 pharmaceutics-10-00277-t001:** Enteric and hepatic transporters of the solute carrier (SLC) superfamily that are targeted by food bioactive compounds.

Transport Enzyme	Food Bioactive Compound	Reference
OATP1A2	- grapefruit compounds	- [24,27,28,30,31]
	- naringin, hesperidin	- [25]
	- apigenin, kaempferol, quercetin	- [32]
OATP1B1	- naringenin, naringin, quercetin, daidzein, genistein, glycyrrhizin, caffeine	- [11]
OATP1B3	- apigenin, kaempferol, quercetin, naringenin, naringin, rutin	- [33]
	- naringenin, naringin, quercetin, daidzein, genistein, glycyrrhizin, caffeine	- [11]
OATP2B1	- grapefruit, orange, and apple compounds	- [26,27,28,29,31]
	- apigenin, kaempferol, quercetin, naringenin, naringin, rutin	- [33]
	- naringenin, naringin, quercetin, daidzein, genistein, glycyrrhizin, caffeine	- [11]
OCT1	- caffeine	- [11]

OATP = Organic Anion Transporting Polypeptides, OCT = Organic Cation Transporters.

**Table 2 pharmaceutics-10-00277-t002:** Enteric and hepatic transporters of the ATP-binding cassette (ABC) superfamily that are targeted by food bioactive compounds.

Transport enzyme	Food bioactive compound	Reference
MDR1	- grapefruit, orange, tangerine, bitter melon compounds, diosmin, nobiletin	- [35,36,37,38,40]
	- Welsh onion compounds	- [36]
	- Ginseng compounds	- [42]
	- sesamin, ginkgolic acid, glycyrrhetinic acid, glabridin	- [43,44]
	- piperine	- [45]
	- carnosic acid	- [46]
	- soybean, dokudami, soymilk, and miso	- [36,47]
	- resveratrol	- [11]
MRP1	- nobiletin	- [40]
	- sulforaphane	- [41]
	- glycyrrhetinic acid	- [43]
MRP2	- grapefruit and orange compounds, naringenin sulforaphane	- [11,35]
	- ginkgo compounds	- [41]
	- apigenin, biochanin A, genistein, glycyrrhizin,	- [44]
	- quercetin, resveratrol	- [11]
BCRP	- bergamottin, tangeretin, nobiletin	- [39]

MDR1 = multidrug resistance protein 1, MRP1 = multidrug resistance-associated protein 1, MRP2 = multidrug resistance-associated protein 2, BCRP = breast cancer resistance protein.

**Table 3 pharmaceutics-10-00277-t003:** Enteric and hepatic metabolizing enzymes of phase I that are targeted by food bioactive compounds.

Metabolizing Enzyme	Food bioactive compound	Reference
CYP1A	- apigenin, apple, caffeine, capsaicin, charcoal-broiled meat, diallyl disulphide, ellagic acid, epicatechin gallate, esperetin, galangin, genistein, grapefruit, hesperetin, isorhamnetin, kaempferol, liquorice compounds, luteolin, naringenin, pinocembrin, piperine, quercetin, sulforaphane, tangeretin, teophillyne	- [11,24,50,51,58,60,61,63,64,65,68,69,70,71,73]
CYP1B	- hesperetin	- [69]
CYP2A	- bergapten, capsaicin, ellagic acid, furanocoumarin, genistein, quercetin, xanthotoxin	- [11,50,67,71]
CYP2B	- bergapten, capsaicin, diallyl disulphide, ellagic acid, furanocoumarin, piperine, xanthotoxin	- [11,64,67]
CYP2C	- apigenin, bergapten, biochanin A, caffeic acid, caffeine, capsaicin, casticin, cranberry, curcumin, daidzein, ellagic acid, galangin, genistein, hesperetin, kaempferol, luteolin, myricetin, naringenin, nicotine, pinocembrin, quercetin, resveratrol, sesamin, xanthotoxin.	- [11,52,54,63,67,74]
CYP2D	- 6-shogaol, apigenin, bergapten, β-sitosterol, capsaicin, curcumin, daidzein, formononetin, galangin, harmine, hesperetin, kaempferol, luteolin, myricetin, pinocembrin, quercetin, resveratrol, serotonin, sesamin, xanthotoxin	- [11]
CYP2E	- diallyl disulphide, ethanol, lycopene, piperine, resveratrol, watercress compounds	- [51,53,64,66,72,75]
CYP3A	- 6-shogaol, apigenin, bergamottin, bergapten, biochanin A, black mulberry, caffeic acid, caffeine, capsaicin, casticin, curcumin, daidzein, diallyl disulphide, diosgenin, ellagic acid, estragole, galangin, epicatechin gallate, genistein, geraniol, glycyrrhizin, harmine, hesperetin, imperatorin, isopimpinellin, kaempferol, luteolin, myricetin, naringenin, naringin, piperine, pomegranate, pyrocatechol, quercetin, resveratrol, rutoside, safrole, sesamin, sulforaphane, tangeretin, taxifolin, theobromine, xanthotoxin	- [4,11,24,28,45,47,48,49,52,54,56,57,62,63,64,67]

CYP = Cytochrome P450 enzyme.

**Table 4 pharmaceutics-10-00277-t004:** Enteric and hepatic metabolizing enzymes of phase II that are targeted by foods or relative compounds.

Metabolizing Enzyme	Food/Bioactive Compound	Reference
SULT	- punicalagin	- [78]
	- sulforaphane, asparagus, eggplant, cauliflower, celery,carrot, and potato compounds	- [80]
UGT	- garden cress and mustard	- [76]
	- milk thistle, epicatechin gallate, saw palmetto,cranberry, quercetin, kaempferol	- [77]
	- (+)-catechin, apigenin, ergosterol, galangin, menthol, naringenin, carvacrol diosgenin, eugenol linalool naringin quercetin, tea polyphenol, borneol citronellol ethinyl estradiol shikimic acid, caproic acid	- [11]
DT-diaphorase	- watercress, garden cress, mustard	- [76]
	- sulforaphane	- [79]
EPH	- soya bean, Brussels sprouts, cauliflower, and onion	- [81]
GST	- sulforaphane	- [79]
	- soya bean, Brussels sprouts, cauliflower, onion	- [81]
NAT	- quercetin	- [50]
	- mustard	- [76]

SULT = Sulfotransferases, UGT = UDP-glucuronosyltransferases, DT-diaphorase = NAD(P)H:quinone oxidoreductase, EPH = Epoxide Hydrolases, GST = Glutathione S-transferases, NAT = *N*-acetyltransferases.

**Table 5 pharmaceutics-10-00277-t005:** Types of food-drug interactions, targets, and food components that can interfere with drug pharmacokinetic/pharmacodynamic profiles.

Types of Food-Drug Interactions	Targets	Food Components
**a. Pharmaceutical**		
	- solubilization, stabilization, and dissolution properties	- alcohol (wine, beer, spirits), capsaicin, cations from milk and dairy products, insoluble fibers (cellulose, bran), soluble fibers (pectin, glucomannan, psyllium, guar gum), lipids, phytoserotonin, proteins
**b. Pharmacokinetic**		
Pre-systemic	- enteric and hepatic transporters of SLC and ABC	- 1-monopalmitin, apigenin, bergamottin, biochanin A, caffeine, carnosic acid, daidzein, genistein, ginkgolic acid, glabridin, glycyrrhetinic acid, glycyrrhizin, hesperidin, kaempferol, naringenin, naringin, nobiletin, piperine, quercetin, resveratrol, rutin, sesamin, soymilk and miso components, sulforaphane, tangeretin, tangeretin
	- enteric and hepatic metabolizing enzymes of CYPP450 families and phase II enzyme reactions, such as SULT, UGT, DT-diaphorase, EPH, GST, NAT	- (+)-catechin, 6-shogaol, apigenin, asparagus components, bergamottin, bergapten, biochanin A, black mulberry components, borneol, caffeic acid, caffeine, caproic acid, capsaicin,, carrot components, carvacrol, casticin, charcoal-broiled meat, citronellol, cranberry components, curcumin, daidzein, diallyl disulphide, dihydrocapsaicin, diosgenin, eggplant components, ellagic acid, epicatechin gallate, ergosterol, estragole, ethanol, ethinyl estradiol, eugenol, formononetin, freeze-dried soya bean components, furanocoumarin, furocoumarin, galangin, genistein, geraniol, glycyrrhizin, harmine, hesperetin, imperatorin, isopimpinellin, isorhamnetin, kaempferol, linalool, luteolin,, lycopene, menthol, milk thistle components, mustard, myricetin, naringenin, naringin, nicotine, onion components, phytoserotonin, pinocembrin, piperine, pomegranate components, potato components, punicalagin, pyrocatechol, quercetin, resveratrol, rutoside, safrole, saw palmetto components, sesamin, shikimic acid, soymilk and miso components, sulforaphane, tangeretin, taxifolin, tea polyphenols, theobromine, theophylline, watercress and garden cress components, xanthotoxin, β-sitosterol
Post-systemic	- transporters, such as albumin, SLC, and ABC. Metabolizing enzymes of CYPP450 families and phase II enzyme reactions, such as UGT	- apigenin, caffeine, casticin, curcumin, genistein, ginseng, isothiocyanates, licorice, lycopene, naringenin, nicotinic acid, proteins, quercetin, resveratrol, resveratrol, rosmarinic acid, salvianolic acid B
**c. Pharmacodynamic**		
	- aromatase, epidermal growth factor, 5-hydroxytryptamine, carbonic anhydrase, estrogen, prostaglandin G/H synthase 2, acetylcholinesterase, cholinesterase, glutamate receptors, D(2) and D(1A), and β2 adrenergic receptors	- α-aminoadipic acid, apigenin, caffeic acid, daidzein, dihydrocapsaicin, eriodictyol, galangin, genistein, glutamate/glutamic acid, harmine, hesperetin, higenamine, kaempferol, liquiritin, luteolin, myricetin, naringenin, P-hydroxybenzoic acid, dopamine, phytoserotonin, quercetin, resveratrol, rutoside, taxifolin

SLC = Solute Carrier, ABC = ATP-Binding Cassette, CYP450 = Cytochrome P450, SULT = Sulfotransferases, UGT = UDP-Glucuronosyltransferases, DT-diaphorase = NAD(P)H:quinone oxidoreductase, EPH = Epoxide Hydrolases, GST = Glutathione S-transferases, NAT = *N*-acetyltransferases

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
