# Peer review of "Food Bioactive Compounds and Their Interference in Drug Pharmacokinetic/Pharmacodynamic Profiles"

_pharmaceutics, 2018, doi:10.3390/pharmaceutics10040277_

Reviewer 1 Report

General comment: The review article entitled “Food Bioactive Compounds and their interference on drug pharmacokinetic/pharmacodynamics profile” presents the interactions between food bioactive compounds and drugs. This is a well-organized review study, with sufficient information in the field.  Some minor corrections are required for the improvement of the manuscript.

Abstract: The Abstract is well written and adequately presents the aim and the basic concept of the study.

Introduction: The introduction section is well-written and explains the aim of the study and the importance to examine food and drug interactions.

-Authors could shortly state that also drugs affect the bioactivity of food components.

-Line 76. The term “methodology of review study” could be stated. Approximately the number of the studies used should be referred.  

Main text:  Authors describe adequately and discuss the possible interactions between bioactive compounds and drugs.

-It is not clear if the majority of the studies in this field are animal experiments or clinical trials. Are there sufficient data from clinical trials? Authors could shortly summarize the basic clinical trials with the design, the results and the proposed mechanism in a Table.

Conclusion: The conclusion is adequate and summarizes the main text.

Bibliography/References: The references used by the authors cover adequately the relative scientific field and the aims of the study.

Comment to the editor-Decision: This is a well-organized study, with sufficient description and discussion of interesting concept.  The manuscript is accepted after minor corrections.

Author Response

Response to Reviewer 1

We would like to thank this Reviewer for his/her very positive remarks about our work, and for the careful and thorough reading of the manuscript. We appreciated his/her constructive comments and suggestions, and we hope that our revised document addresses all his/her important issues satisfactorily.

 Comments and Suggestions for Authors

a.    General comment: The review article entitled “Food Bioactive Compounds and their interference on drug pharmacokinetic/pharmacodynamics profile” presents the interactions between food bioactive compounds and drugs. This is a well-organized review study, with sufficient information in the field.  Some minor corrections are required for the improvement of the manuscript.

b.    Abstract: The Abstract is well written and adequately presents the aim and the basic concept of the study.

c.    Introduction: The introduction section is well-written and explains the aim of the study and the importance to examine food and drug interactions.

               i.     Authors could shortly state that also drugs affect the bioactivity of food components.

             ii.     Line 76. The term “methodology of review study” could be stated. Approximately the number of the studies used should be referred.

d.    Main text:  Authors describe adequately and discuss the possible interactions between bioactive compounds and drugs.

               i.     It is not clear if the majority of the studies in this field are animal experiments or clinical trials. Are there sufficient data from clinical trials? Authors could shortly summarize the basic clinical trials with the design, the results and the proposed mechanism in a Table.

e.    Conclusion: The conclusion is adequate and summarizes the main text.

f.     Bibliography/References: The references used by the authors cover adequately the relative scientific field and the aims of the study.

g.    Comment to the editor-Decision: This is a well-organized study, with sufficient description and discussion of interesting concept.  The manuscript is accepted after minor corrections.

Revisions

c.    Introduction: The introduction section is well-written and explains the aim of the study and the importance to examine food and drug interactions.

           i.     Authors could shortly state that also drugs affect the bioactivity of food components.

Thanks for pointing out this important issue. We agree on the assertion that drugs may affect the bioactivity of certain foods and their components. In fact, the food-drug interaction is definitely bi-directional. However, we decided to focus on the unidirectionality from foods to drugs so as not to divert the less wary reader's attention. Indeed, we agree with the reviewer’s consideration, and we corrected and integrated the sentence at line 576 as follows: “…Food bioactive compounds bioavailability may be influenced by many factors, such as the luminal pH of the intestine [107], inter-individual variabilities, hydrophobic properties of the food molecule [108], or even drugs themselves. In fact, drug metabolism can directly alter the same steps of food components kinetic profile, and some medications may cause gastrointestinal dismotility or diarrhea, thus indirectly affecting food components bioavailability.

         ii.     Line 76. The term “methodology of review study” could be stated. Approximately the number of the studies used should be referred.

We appreciate the positive feedback from the reviewer. We corrected at line 77 as follows: “…1.1. Methodology of the review”. For what concerns the number of studies, we included a variety in the information sources, as for example “…Google Scholar, PubMed, and Scopus databases…”, and provided the reader with a broader and more representative view of the topics. Not being systematic, the exhaustiveness of the arguments was guaranteed by tackling each single narrative step that could be a cornerstone to reach the conclusion. For the lack of a narrow scope, we are unable to detail the number of studies used, but we are confident that the original references have been cited.

d.    Main text:  Authors describe adequately and discuss the possible interactions between bioactive compounds and drugs.

           i.     It is not clear if the majority of the studies in this field are animal experiments or clinical trials. Are there sufficient data from clinical trials? Authors could shortly summarize the basic clinical trials with the design, the results and the proposed mechanism in a Table.

Thanks for these important request for clarification. When we decided how to organize the manuscript, we have been wary to elucidate whether the reported data came from a preclinical or a clinical study. For example, we frequently reported the term in vivo or in vitro when reporting a specific study. Conversely, we clarified if data came from a clinical study on humans. For example, at line 387…In a clinical trial, charcoal-broiled meat induced CYP1A enzymes in the liver and small intestine of healthy volunteers [70]…” and at line 391…In a clinical trial, CYP1A2 activity was concluded to be increased by an enriched-brassica diet, but decreased by an enriched-apiaceous diet (dill, celery, parsley, parsnip - Pastinaca sativa L.) [73]”. Moreover, we reported at line 560 that “…Most of data concerning food interference with drug metabolism comes from in vitro or animal studies, but preclinical models cannot be uniquely relied upon for predicting relevant human food-drug interactions, as large differences among species in the ability to metabolize xenobiotics exist”. At line 574 we continued saying that “…a relevant preclinical interaction may be a non-relevant clinical interaction. For example, few clinical effects were suggested when considering the interference of food components with plasma protein binding [106]”. As a conclusion and in order to better convey the decisive message about clinical trials, we rephrased the sentences at line 586 as follows: “…Clinical trials showed significant pharmacokinetic interactions, whose practical consequences on pharmacological response showed to vary. Of note, only grapefruit juice appeared to give the most promising clinical data, probably due to its now-dated interest. However, fruits from Citrus genus, such as Seville orange or pomelos, may have the same effect as grapefruit juice and their consumption should be avoided if taking medicines that interact with grapefruit juice. Still, to study other suggested food-drug interactions in humans clinical trial would be better in order to prevent them before they would be observed in daily practice”. For what concerns the proposed mechanism, we agree with the reviewer that it would be fair to explain the underlying mechanisms for which a single clinical effect is explainable and traceable to a specific biological effect. However, we started presenting literature results following the metabolic pathway of food-drug interactions - pharmaceutical, pharmacokinetic, and then pharmacodynamic - and we have guaranteed the explanation of the mechanisms underlying a priori a specific clinical effect. In fact, at line 115 we suggested that clinical observations of pharmacodynamic interactions“…could be considered the indirect result of previous pharmaceutical or pharmacokinetic interactions that could ultimately alter the clinical effect of the drug”. Moreover, giving previous corrections concerning relevant clinical trials, we may conclude that only grapefruit juice appeared to give the most promising results from clinical trials, and that its biological mechanisms have been extensively explained in the text.

Reviewer 2 Report

In this review manuscript, the authors extensively cited literature to give a summary of the interactions between drugs and food components. The authors started with a reasonable classification for these interactions and most of the major points in this topic have been covered. However, it seems that the major attention is still paid to the pharmacokinetics interactions part, and the pharmacodynamics part is inappropriately underemphasized. It should be pointed out that the PD interactions between food components and drug target protein are not only derived from bioinformatic evidence but also extensively observed in preclinical and clinical studies. (Some drugs are actually derived from efficacious natural products.)

A major suggestion is to use tables containing food name, components name, enzyme/transporter, and reference information, instead of long descriptive paragraphs. Inside each section, giving a lot of examples in words make it difficult to read.

Page 4, line 155-173. It should be noted that the efficacy of a drug can be AUC-driven or Cmax-driven. Thus the changes of AUC or Tmax may have different effects on drug efficacies.

Page 5, line 216-232. The predominant role of CYPs in phase I metabolism is of course well-recognized. However, some other enzymes like FMO, ADH, and MAO should also be mentioned in this paragraph.

Page 10, line 470-473. Here “transporters and metabolizing enzymes” should be referred to as drug targets. The pharmacokinetics interactions have already been discussed in the previous sections.

Author Response

Response to Reviewer 2

We would like to thank the Reviewer 2 for careful and thorough assessment of our manuscript and for the attentive comments and constructive suggestions. We have tried to comply with his/her requests and hope that our corrections are satisfactory. A native English speaker corrected the manuscript for what concerns grammar and syntax. Our detailed responses to each suggestion are listed below.

Comments and Suggestions for Authors

a.    In this review manuscript, the authors extensively cited literature to give a summary of the interactions between drugs and food components. The authors started with a reasonable classification for these interactions and most of the major points in this topic have been covered.

b.    However, it seems that the major attention is still paid to the pharmacokinetics interactions part, and the pharmacodynamics part is inappropriately underemphasized. It should be pointed out that the PD interactions between food components and drug target protein are not only derived from bioinformatic evidence but also extensively observed in preclinical and clinical studies. (Some drugs are actually derived from efficacious natural products.)

c.    A major suggestion is to use tables containing food name, components name, enzyme/transporter, and reference information, instead of long descriptive paragraphs. Inside each section, giving a lot of examples in words make it difficult to read.

d.    Page 4, line 155-173. It should be noted that the efficacy of a drug can be AUC-driven or Cmax-driven. Thus the changes of AUC or Tmax may have different effects on drug efficacies.

e.    Page 5, line 216-232. The predominant role of CYPs in phase I metabolism is of course well-recognized. However, some other enzymes like FMO, ADH, and MAO should also be mentioned in this paragraph.

f.     Page 10, line 470-473. Here “transporters and metabolizing enzymes” should be referred to as drug targets. The pharmacokinetics interactions have already been discussed in the previous sections.

Revisions

b.    However, it seems that the major attention is still paid to the pharmacokinetics interactions part, and the pharmacodynamics part is inappropriately underemphasized. It should be pointed out that the PD interactions between food components and drug target protein are not only derived from bioinformatic evidence but also extensively observed in preclinical and clinical studies. (Some drugs are actually derived from efficacious natural products.)

Thanks for highlighting this important issue: we agree with your comment. In fact, pharmacodynamic interactions derive also from clinical studies that investigated the synergistic effects of natural products and drugs. We pointed out and integrated the text at line 493 as follows: “A variety of pharmacodynamic interactions was observed in preclinical, clinical, but also bioinformatic studies, which consider the predicted bioactivity profile, as the chemical space of food components was linked with drug target space. Foods that contained the most interfering components with drug targets, such as receptors, resulted to be…”.

c.    A major suggestion is to use tables containing food name, components name, enzyme/transporter, and reference information, instead of long descriptive paragraphs. Inside each section, giving a lot of examples in words make it difficult to read.

We agree with your statement regarding the long descriptive paragraphs. However, it is necessary to define the contexts of the experiments or trials, which would be difficult to report and assume in a table. Furthermore, we thought that the conclusive table could be sufficient to provide a broad and easily comprehensible vision. Indeed, even if it strongly changes the structure of the manuscript, we support the major suggestion: according to your proposal, we have included a summary table for the most wordy paragraphs. Table 1, table 2, table 3, and table 4 were introduced in the text, regarding the enteric/hepatic transporters and the enteric/hepatic metabolizing enzymes.

d.    Page 4, line 155-173. It should be noted that the efficacy of a drug can be AUC-driven or Cmax-driven. Thus the changes of AUC or Tmax may have different effects on drug efficacies.

Thanks for these important request for clarification. The efficacy of some drugs may indeed be based on its pharmacokinetic profile, thus being, for example, Cmax-driven. To fill this gap in the text, we have modified and added the following information at line 163: “If food delays the onset of drug action, the efficacy might not be affected, despite differences in the pharmacokinetic profile. Conversely, the efficacy of some drugs, such as sildenafil, may be Cmax-driven and the delay of Tmax due to the concomitant food consumption can affect clinical efficacy. In fact, when food intake causes a significant change in the AUC of a drug, the overall amount absorbed could be reduced and would cause a far more clinical effect.

e.    Page 5, line 216-232. The predominant role of CYPs in phase I metabolism is of course well-recognized. However, some other enzymes like FMO, ADH, and MAO should also be mentioned in this paragraph.

We appreciate the positive feedback from the reviewer. We corrected at line 232 as follows: “…Phase I reactions are also mediated by Flavin-containing monooxygenases (FMO). Other oxidative reactions are catalyzed by alcohol dehydrogenases (ADH) and Monoamine Oxidases (MAO), respectively localized in the cytosol and on mitochondrial outer membrane of the liver. Phase II reactions involve other metabolizing or conjugating enzymes, such as Sulfotransferases (SULT), UDP-glucuronosyltransferases (UGT), NAD(P)H:quinone oxidoreductase (DT-diaphorase), Epoxide Hydrolases (EPH), Glutathione S-transferases (GST) and N-acetyltransferases (NAT), all of which produce end products more readily excreted. The conjugation reactions, by promoting excretion in the bile and/or the urine, affect detoxification”.

f.     Page 10, line 470-473. Here “transporters and metabolizing enzymes” should be referred to as drug targets. The pharmacokinetics interactions have already been discussed in the previous sections.

At line 495, we corrected “…Foods that contained the most interfering components with drug targets, such as receptors, resulted to be…
